# Functional Characterization of the Solute Carrier LAT-1 (SLC7A5/SLC2A3) in Human Brain Capillary Endothelial Cells with Rapid UPLC-MS/MS Quantification of Intracellular Isotopically Labelled L-Leucine

**DOI:** 10.3390/ijms23073637

**Published:** 2022-03-26

**Authors:** Cindy Bay, Gzona Bajraktari-Sylejmani, Walter E. Haefeli, Jürgen Burhenne, Johanna Weiss, Max Sauter

**Affiliations:** Department of Clinical Pharmacology and Pharmacoepidemiology, Heidelberg University Hospital, Im Neuenheimer Feld 410, 69120 Heidelberg, Germany; cindy.bay@med.uni-heidelberg.de (C.B.); gzona.bajraktari-sylejmani@med.uni-heidelberg.de (G.B.-S.); walter-emil.haefeli@med.uni-heidelberg.de (W.E.H.); juergen.burhenne@med.uni-heidelberg.de (J.B.); johanna.weiss@med.uni-heidelberg.de (J.W.)

**Keywords:** LAT-1, amino acid transport, CNS, blood–brain barrier, tandem mass spectrometry, UPLC, L-leucine

## Abstract

The solute carrier L-type amino acid transporter 1 (LAT-1/SLC7A5) is a viable target for drug delivery to the central nervous system (CNS) and tumors due to its high abundance at the blood–brain barrier and in tumor tissue. LAT-1 is only localized on the cell surface as a heterodimer with CD98, which is not required for transporter function. To support future CNS drug-delivery development based on LAT-1 targeting, we established an ultra-performance liquid chromatography–tandem mass spectrometry (UPLC-MS/MS) assay for stable isotopically labeled leucine ([^13^C_6_, ^15^N]-L-leucine), with a dynamic range of 0.1–1000 ng/mL that can be applied for the functional testing of LAT-1 activity when combined with specific inhibitors and, consequently, the LAT-1 inhibition capacity of new compounds. The assay was established in a 96-well format, facilitating high-throughput experiments, and, hence, can support the screening for novel inhibitors. Applicable recommendations of the US Food and Drug Administration and European Medicines Agency for bioanalytical method validation were followed to validate the assay. The assay was applied to investigate the IC_50_ of two well-known LAT-1 inhibitors on hCMEC/D3 cells: the highly specific LAT-1 inhibitor JPH203, which was also used to demonstrate LAT-1 specific uptake, and the general system L inhibitor BCH. In addition, the [^13^C_6_, ^15^N]-L-leucine uptake was determined on two human brain capillary endothelial cell lines (NKIM-6 and hCMEC/D3), which were characterized for their expressional differences of LAT-1 at the protein and mRNA level and the surface amount of CD98. The IC_50_ values of the inhibitors were in concordance with previously reported values. Furthermore, the [^13^C_6_, ^15^N]-L-leucine uptake was significantly higher in hCMEC/D3 cells compared to NKIM-6 cells, which correlated with higher expression of LAT-1 and a higher surface amount of CD98. Therefore, the UPLC-MS/MS quantification of ([^13^C_6_, ^15^N]-L-leucine is a feasible strategy for the functional characterization of LAT-1 activity in cells or tissue.

## 1. Introduction

The central nervous system (CNS) is shielded from the systemic circulation by the blood–brain barrier (BBB), which prevents the penetration of xenobiotics, including the majority of drugs. On the other hand, the BBB is permeable for nutrients such as amino acids, which are actively transported into the CNS by amino acid transporters. Member 5 of subfamily A of the solute carrier family 7 (SLC7A5), also known as L-type amino acid transporter 1 (LAT-1), shuttles large and neutral amino acids into the cell [1]. LAT-1 (SLC7A5) was originally identified as a heterodimer partner of CD98 (SLC2A3, also known as 4F2hc), connected via a disulfide bond [1,2]. LAT-1 mediates the transport of amino acids; however, the formation of the CD98-heterodimer is required for the localization of LAT-1 to the cell surface [3], and CD98 may also contribute to the transport function at the cell surface membrane, although this finding is still the subject of ongoing discussion [4,5]. Amino acids are recognized by their amino and carboxyl function and require a large neutral side chain to be the substrate of LAT-1 [6]. A comprehensive overview of the transport function, structure, and role of CD98-heterodimerization of LAT-1 can be found elsewhere [7,8]. In the following, LAT-1_HD_ will be used to describe the functional heterodimer, while SLC7A5/LAT-1 and SLC2A3/CD98 denote the individual partners. Powered by the antiport of abundant intracellular amino acids, LAT-1_HD_ contributes to the supply of essential amino acids, including L-leucine [9]. As a consequence, LAT-1_HD_ is dependent on the uptake of amino acids by other amino acid transporters. The information on the complex interplay of amino acid transporters has been summarized elsewhere [10].

The BBB is primarily formed by brain capillary endothelial cells, where LAT-1 is highly expressed [6,11,12,13], and likely plays an important role in the supply to the CNS of branched-chain amino acids, including L-leucine [9,14]. The expression of LAT-1_HD_ in these cells not only enables the supply of essential amino acids to the brain [15] but can also be used for the uptake of amino acid-like drugs. In fact, LAT-1_HD_ was identified as the responsible transporter for the brain accessibility of a number of amino acid-like drugs, including L-DOPA [12]. Therefore, LAT-1_HD_ is of considerable interest for targeted delivery to the brain [6].

Furthermore, LAT-1_HD_ is reportedly overexpressed in various cancer entities, which has been reviewed elsewhere [8,16], including glioblastoma [17], thus covering the increased nutritional demand necessary for cell growth [8,18], and its expression correlates with increased angiogenesis and metastasis [19]. High LAT-1_HD_ expression at the BBB and in brain tumors can be used for the PET imaging of tumors with radiolabeled amino acid tracers [20,21,22] and as a potential target for cancer pharmacotherapy. Various LAT-1_HD_ targeting approaches for cancer treatment and brain delivery are currently under investigation [23,24,25,26,27]. A potential dual targeting of brain tumors, based on the high expression of LAT-1_HD_ in both the BBB and brain tumors, could contribute to the development of effective treatment options. Therefore, a fast and reliable functional assay for LAT-1_HD_ could be of great value for the drug development of new and specific inhibitors and substrates of this transporter. Due to its high affinity for LAT-1_HD_, L-leucine has emerged as the gold standard for LAT-1_HD_ uptake investigations. Because of the endogenous background of L-leucine in biological systems, a direct measurement of L-leucine uptake by amino acid transporters either requires overexpression of these transporters to increase the signal-to-noise ratio of the uptake, accounting for this background, or the use of isotopologes. Direct measurement of amino acid uptake can especially be realized in heterologous expression systems with low intracellular levels of amino acids, such as Xenopus laevis oocytes [28]. Although ^13^C-isotopologes of amino acids have been used for uptake experiments [10], especially in the context of metabolism studies [29], previous investigations of L-leucine uptake in the context of transporter characterization were primarily based on the radioactive isotopologes [^14^C]-L-leucine or [^3^H]-L-leucine [5,10,30,31,32,33,34,35,36,37]. Nevertheless, through comprehensive uptake studies with different amino acids and the consideration of the respective metabolism, the specific contribution of LAT-1_HD_ for L-leucine uptake and, hence, functional characterization can be performed [10]. However, this sophisticated methodology requires extensive time and experimental efforts.

In this work, we have developed an ultra-performance liquid chromatography—tandem mass spectrometry (UPLC-MS/MS) assay for the quantification of isotopically labeled [^13^C_6_, ^15^N]-L-leucine for the functional characterization of LAT-1_HD_ using [^2^H_3_]-L-leucine as an internal standard (IS). To demonstrate the reliability of our measurements, we validated the assay following the applicable sections of the guidelines for bioanalytical method validation of the US Food and Drug Administration (FDA) and European Medicines Agency (EMA) [38,39]. We adapted a previously developed approach for semi-high throughput sample processing in a 96-well format [40] and further reduced the necessary homogenate volume to 25 µL. The absence of sample transfer steps and the use of an isotopologe as IS enabled fast and highly accurate [^13^C_6_, ^15^N]-L-leucine quantification. We used the assay to determine the IC_50_ values of two known competitive LAT-1_HD_ inhibitors with large differences in potency: the highly specific LAT-1_HD_ inhibitor, JPH203 [31], which was also used to assess the contribution of LAT-1_HD_ to [^13^C_6_, ^15^N]-L-leucine uptake, and a general system L inhibitor, 2-aminobicyclo-(2,2,1)-heptane-2-carboxylic acid (BCH) [41,42]. While JPH203 is not transported [43], BCH is a LAT substrate [32,41]. The K_i_ value of JPH203 for LAT-1_HD_ is 0.058 µM [31] and three orders of magnitude higher for BCH (50.0–55.1 µM) [41,44]. 

Finally, LAT-1_HD_ uptake activity in two immortalized human brain capillary endothelial cell lines was investigated to evaluate their suitability for the screening of drugs and inhibitors targeting LAT-1_HD_. For this purpose, the commercially available human brain capillary microvascular endothelial cell-line hCMEC/D3 [45] and the in-house developed human brain capillary endothelial cell-line NKIM-6 [46] were compared. Expressional differences of LAT-1 and CD98 between the cell lines were determined, and the functional characterization of LAT-1_HD_ was evaluated by [^13^C_6_, ^15^N]-L-leucine uptake and inhibition with the LAT-1 selective inhibitor JPH203 [31].

## 2. Results

### 2.1. Method Development

Positive ESI efficiently generated the [M + H]^+^ ion of [^13^C_6_, ^15^N]-L-leucine (*m/z* 139.0). Optimization of collision-induced dissociation (CID) dissociation yielded the iminium ion of [^13^C_6_, ^15^N]-L-leucine at *m/z* 92.0, by far the most abundant product ion (Figure 1 and Appendix A). Only one other product ion was observed at applicable intensity for sensitive selected reaction monitoring (SRM) at *m/z* 120.0. This product ion was generated at slightly higher collision energy than the iminium ion and corresponds to the loss of water from the precursor ion (Appendix A). Because of the superior intensity, the mass transition of *m/z* 139.0 → 92.0 was used for SRM. For the IS, the corresponding mass transition of *m/z* 135.0 → 89.0 was monitored. Figure 1 depicts the product spectrum and structure of [^13^C_6_, ^15^N]-L-leucine and the monitored fragment, including the positions of isotopic labels.

The chromatography of [^13^C_6_, ^15^N]-L-leucine and IS [^2^H_3_]-L-leucine was disturbed by endogenous compounds. As a consequence, efficient separation was required. We aimed to use hydrophilic interaction liquid chromatography (HILIC) due to its compatibility with our planned sample processing procedure. On a pure silica HILIC column (Waters Cortecs HILIC UPLC^®^ column [1.6 µm, 2.1 mm × 50 mm]) and mobile phases with HCOOH, insufficient retention of [^13^C_6_, ^15^N]-L-leucine was achieved. While mobile phases with ammonium formiate resulted in good retention, the resulting peaks were broad, regardless of the used gradient. In contrast, a Waters BEH Amide UPLC^®^ column (130 Å, 1.7 µm, 2.1 mm × 50 mm) produced sharp peaks and efficient chromatographic separation using flat gradients. Again, retention was considerably higher with mobile phases containing ammonium formiate. A very flat gradient from 0 to 8% aqueous eluent within 2.4 min was found optimal for efficient separation of the endogenous interference (Figure 2C) and resulted in sharp peaks of 6 s at baseline. Due to the high initial retention on the BEH amide column, a high injection volume of 20 µL was found applicable for the assay, which facilitates sensitive measurements. The column flushing step after the elution of [^13^C_6_, ^15^N]-L-leucine was kept short, which was considered sufficient for a UPLC separation. While this might slightly reduce column lifetime, sample throughput was increased.

For sample preparation, we aimed at using the 96-well plates used for cell culture for the complete sample preparation procedure and injection onto the UPLC-MS/MS system to circumvent the need for sample transfer steps to achieve high sample throughput, as previously described [40]. As a consequence, solely protein precipitation is feasible for analyte extraction. Due to the compatibility with HILIC chromatography, we performed this with acetonitrile, which resulted in excellent recovery (Table 1).

Representative UPLC-MS/MS chromatograms of blank samples, samples containing only IS, LLOQ, mid quality control (QC), and a study sample (10 min incubation with 10 µM [^13^C_6_, ^15^N]-L-leucine and 100 µM JPH203) are shown in Figure 2.

### 2.2. Validation of the UPLC-MS/MS Quantification

After method development, we validated the [^13^C_6_, ^15^N]-L-leucine quantification assay for a dynamic range of 4 orders of magnitude from 0.1 to 1000 ng/mL according to FDA and EMA guidelines for bioanalytical method validation [38,39] with regard to interday and intraday accuracy and precision, linearity, extract stability, specificity, carry-over, recovery, and matrix effect. Validation was performed with calibration and QC samples prepared by spiking cell lysates. In all calibration curves, a correlation coefficient (r^2^) > 0.99 was accomplished for linear regression (Appendix A and Appendix A). Intraday and interday accuracy and precision of the quantified QC samples at LLOQ, low, mid, and high QC concentrations are listed in Table 2 and well complied with the required limits of the guidelines (100 ± 15% [20% at LLOQ] for accuracy and ≤15% coefficient of variation [CV] [20 % at LLOQ] for precision).

The selectivity of the assay was demonstrated with blank cell lysate samples, in which no interference was detected at the retention time of [^13^C_6_, ^15^N]-L-leucine. In addition, no carry-over was observed after blanks queued after the highest calibration samples. Finally, recovery was very high and consistent across QC concentrations, and the IS and IS-normalized matrix effect fully complied with the required limits of 100 ± 15 % (Table 2), which demonstrates the capability of the IS to efficiently balances these effects. As a prerequisite for the analysis of multiple queued samples, [^13^C_6_, ^15^N]-L-leucine was found stable in the extracts at the autosampler temperature of 10 °C for at least 24 h. This was demonstrated by the accurate quantification of stored QC samples with freshly prepared calibration samples (accuracy ranging from 98.8% to 107.0% for low, mid, and high QC; Appendix A).

### 2.3. Determination of IC_50_

The developed LAT-1_HD_ assay was subsequently used to determine the inhibitory capacity of two competitive inhibitors on hCMEC/D3 cells. The first, JPH203, is currently studied in a clinical trial and shows high specificity for LAT-1_HD_ [24,31]. The second, BCH, is a non-selective inhibitor for L-system amino acid transporters and a substrate of LAT-1 [30], LAT-2 [42], LAT-3 [47], and LAT-4 [48], as well as other amino acid transporters such as B^0^AT2 [10]. Based on previously reported IC_50_ values [30,31,32,44], the investigated concentration ranges for the two inhibitors were selected. 

Cells were incubated for 10 min with the respective inhibitor concentration and 10 µM [^13^C_6_, ^15^N]-L-leucine. L-leucine is the gold-standard amino acid widely used to characterize LAT-1_HD_ [5,30,31,32,33,34,35,36]. IC_50_ was determined with three independent experiments with at least three replicates. 

A concentration-dependent inhibition of [^13^C_6_, ^15^N]-L-leucine uptake was observed for JPH203 and BCH (Figure 3). The maximal inhibition was considerably higher for the non-selective inhibitor BCH. The IC_50_ value for JPH203 was 0.103 ± 0.015 µM and 91 ± 39 µM for BCH. Because IC_50_ values depend on the chosen experimental parameters, including substrate concentration, we calculated the respective K_i_ values, which are intrinsic thermodynamic values of the transporter-inhibitor interaction and, therefore, more comparable. This was performed using the IC-50-to-K_i_ web-based tool [49]. For the calculation of K_i_ values, we assumed competitive inhibition and a K_m_ for L-leucine of 32 µM [1,2,50], which allows us to derive the K_i_ according to the Cheng-Prusoff equation: $K_{i}=\frac{IC_{50}}{\frac{\left[S\right]}{K_{m}}+1}.$ This resulted in Ki values of 0.082 and 72.8 µM for JPH203 and BCH, respectively. The reduction of [^13^C_6_, ^15^N]-L-leucine uptake was substantially higher for inhibition with BCH compared to JPH203, with a total reduction of 96% and 80%, respectively. The absolute baseline uptake (bottom plateau of the inhibition curve) under BCH inhibition corresponded to 1.6 pmol/mg protein compared to 8.7 pmol/mg protein for the LAT-1 specific inhibitor JPH203.

### 2.4. [^13^C_6_, ^15^N]-L-leucine uptake in hCMEC/D3 and NKIM-6 cells

LAT-1_HD_ activity in two different human brain capillary cell lines was characterized by measuring the uptake of [^13^C_6_, ^15^N]-L-leucine with our validated UPLC-MS/MS assay. To correct for different cell numbers between the experiments, intracellular [^13^C_6_, ^15^N]-L-leucine concentration was normalized to protein content measured in at least 4 wells distributed across the 96-well plate. With a mean of 19.3 ± 3.9 µg protein/well, the protein amount was stable between experiments, cell passages, and different cell types. The standard deviation (SD) between the protein content replicates on individual plates was ≤1.3 µg/well, which corresponds to a variation of <6.8%. The mean for hCMEC/D3 cells accounted for 18.8 ± 4.0 µg/well and for NKIM-6 20.7 ± 3.0 µg/well. The uptake of increasing concentrations of [^13^C_6_, ^15^N]-L-leucine was determined without inhibitor and in the presence of 5 µM JPH203 and 500 µM BCH in both cell lines (Figure 4). The incubation time for the amino acid transporter was set at 10 min because this was the shortest time that was well-manageable. Substances were diluted in pre-warmed buffer. 

As depicted in Figure 4, a concentration-dependent increase in [^13^C_6_, ^15^N]-L-leucine uptake was observed. The addition of 5 µM JPH203 and 500 µM BCH decreased this uptake. However, in the highest [^13^C_6_, ^15^N]-L-leucine concentration, inhibition by BCH was considerably less efficient compared to the lower concentrations. Because of the higher K_i_ of BCH in comparison to [^13^C_6_, ^15^N]-L-leucine, BCH is not capable of efficiently inhibiting [^13^C_6_, ^15^N]-L-leucine uptake if not present in substantial excess. This leads to a curvature of the relationship because the increasing [^13^C_6_, ^15^N]-L-leucine concentration approaches that of BCH.

Overall, NKIM-6 showed substantially lower [^13^C_6_, ^15^N]-L-leucine uptake (Figure 4B). Further, uptake of [^13^C_6_, ^15^N]-L-leucine seemed to approach saturation in the highest concentration on hCMEC/D3 cells (Figure 4A). Additionally, the predominant fraction of the [^13^C_6_, ^15^N]-L-leucine uptake at 1 µM was inhibited by JPH203 on hCMEC/D3 cells (reduction of 77 %), while on NKIM-6 cells, JPH203 only inhibited less than half of the L-leucine uptake (reduction of 45%).

### 2.5. Quantification of SLC7A5 and SCL2A3 Expression

To verify that the lower substrate uptake observed in NKIM-6 is indeed due to a lower LAT-1_HD_ expression compared to hCMEC/D3, we evaluated the protein and mRNA content in both cell lines. As shown in Figure 5A, *SLC7A5* mRNA expression was significantly higher in hCMEC/D3 compared to NKIM-6.

Because only the heterodimer LAT-1_HD_ (SLC7A5/SLC2A3) is functional, we draw several lines of evidence to show the higher abundance of functional transporter protein in hCMEC/D3 cells. First, we quantified the main extracellular part of the heterodimer, SLC2A3/CD98, on the cell surface by flow cytometry. As shown in Figure 5B, hCMEC/D3 cells demonstrated significantly higher median fluorescence intensity corresponding to a higher CD98 amount on the cell surface.

Because CD98 also dimerizes with other transporters [42], SLC7A5/LAT-1 was further quantified with SDS-PAGE and Western blot (Figure 5C,D). SLC7A5/LAT-1 was considerably expressed in both cell lines. The intensity of the LAT-1 bands was significantly higher in hCMEC/D3 protein lysates. In combination with the flow cytometry data, these results indicate a higher abundance of functional LAT-1_HD_ in hCMEC/D3 cells compared to NKIM-6 cells. 

## 3. Discussion

Due to its elevated abundance in the BBB and in various tumor malignancies, including glioblastoma, LAT-1_HD_ is a viable target for CNS and tumor delivery via prodrug or drug delivery system approaches or therapeutically used inhibitors. Because it is highly expressed in brain tumors and due to its importance at the BBB, LAT-1_HD_ is of particular interest for currently difficult-to-treat brain tumors. Therefore, rapid screening of inhibitors can support efficient and innovative drug development. Because all substrates are competitive inhibitors of the transporter, screening of inhibitory characteristics is a viable first step in substrate or inhibitor identification. Due to the fact that L-leucine is the gold-standard substrate for the characterization of LAT-1_HD_ function, the determination of L-leucine uptake is a promising approach to inhibitor screening. In the presence of the endogenous occurrence of L-leucine, such investigations may be carried out with isotopologes. Up to date, this was mainly performed with radioactive-labeled L-leucine (^3^H and ^14^C). However, these low-energy β-emitters are potentially harmful, have a long decay time (t_1/2_ of 12.3 and 5730 a for ^3^H and ^14^C, respectively), and require laborious sample processing to enable their measurement by scintillation. It is, therefore, desirable to circumvent the use of radioactive tracers whenever possible.

To the best of our knowledge, our developed quantification of [^13^C_6_, ^15^N]-L-leucine is the first established LAT-1_HD_ functional UPLC-MS/MS assay based on L-leucine using different isotopologes, which facilitates the accurate quantification of L-leucine uptake due to the identical physicochemical characteristics of analyte and IS. The endogenous presence of high amounts of L-leucine in cell homogenates impedes the accurate measurement of L-leucine concentrations for LAT-1_HD_ characterization. However, stable isotopically labeled L-leucine, especially ^13^C and ^15^N labeled analogs, do accurately reflect L-leucine uptake (due to their identical physicochemical characteristics). As a consequence, the use of differently labeled isotopologes of L-leucine is a viable approach to direct measurement of cellular L-leucine uptake with UPLC-MS/MS. Because L-leucine can be taken up by other amino acid transporters, LAT-1_HD_ contribution to the uptake needs to be assessed by LAT-1 specific inhibition, which can be performed with the specific inhibitor JPH203 [31].

The developed assay is considered a semi-high throughput method due to sequential sample quantification by the UPLC-MS/MS system, allowing for the measurement of L-leucine uptake in a 96-well format. The assay met all applicable requirements for bioanalytic method development of the FDA and EMA, demonstrating its reliability. Due to its high sensitivity, quantification can be performed in 96-well formats and with a rapid sample processing strategy directly in the cell culture wells, which is highly beneficial for screening experiments. The wide dynamic range of over 4 orders of magnitude (between 0.1 ng/mL to 1000 ng/mL) is advantageous for a variety of investigations.

The applicability of our assay was demonstrated with the determination of IC_50_ values for two well-described LAT-1_HD_ inhibitors: JPH203, a highly selective, potent LAT-1_HD_ inhibitor, and BCH, a non-selective LAT inhibitor with low potency. The determined IC_50_ values were in concordance with the literature: 0.103 ± 0.015 µM for JPH203 (vs. 0.06 µM [31]) and 91 ± 39 µM for BCH (vs. 73.1–131.5 µM [30,32,44]). Moreover, the K_i_ values for both inhibitors were in reasonable agreement with previously reported values for the inhibition of LAT-1_HD_-mediated L-leucine uptake: 0.082 µM for JPH203 (vs. 0.058 µM as calculated based on data of Oda and coworkers [31]) and 72.8 µM for BCH (vs. 50.0–55.1 µM [41,44]).

Further, our developed assay can be used to assess the activity of LAT-1_HD_ in different cells (or tissues). To demonstrate the LAT-1_HD_ specificity of the [^13^C_6_, ^15^N]-L-leucine uptake, inhibition with JPH203 can be used to evaluate the LAT-1_HD_ dependent fraction. We compared two brain capillary endothelial cell lines for their LAT-1_HD_ function to evaluate their suitability for experiments on drug delivery across the BBB: hCMEC/D3 and NKIM-6 cells. The determined levels of functional LAT-1_HD_ were substantially higher in hCMEC/D3 cells compared to NKIM-6 cells. These results were consistent for leucine uptake, mRNA levels of SLC7A5/LAT-1, protein levels of SLC7A5/LAT-1, and surface amount of SLC3A2/CD98. The used cell lines are known to form monolayers with contact inhibition, preventing overgrowth [45,46]. This was reflected in the high consistency of protein content in individual and between 96-well plates. This consistency in protein content and, hence, cell numbers per well facilitates reliable comparisons of individual experiments.

In the IC_50_ determinations on hCMEC/D3 cells, the baseline uptake of [^13^C_6_, ^15^N]-L-leucine was considerably lower in the presence of BCH compared to JPH203. This is in accordance with the non-selective inhibitory effect of BCH for the LAT system [30,42,47,48] and other transporters such as B^0^AT2 [10], which may contribute to baseline uptake of [^13^C_6_, ^15^N]-L-leucine. Because the LAT-1 specific inhibitor JPH203 inhibits the majority of [^13^C_6_, ^15^N]-L-leucine uptake in hCMEC/D3 cells, it can be concluded that LAT-1_HD_ is the primary transporter responsible for L-leucine uptake in these brain capillary endothelial cells. Even close to physiological relevant L-leucine concentrations (150–200 µM [51]) of 100 µM, the LAT-1_HD_-dependent fraction of [^13^C_6_, ^15^N]-L-leucine uptake on hCMEC/D3 cells is still almost 70%, which indicates a predominant role of LAT-1_HD_ for L-leucine uptake if this finding translates to the brain capillary endothelial cells and BBB in vivo remains elusive. In contrast to hCMEC/D3 cells, [^13^C_6_, ^15^N]-L-leucine uptake was not primarily dependent on LAT-1_HD_ on NKIM-6 cells at a similar JPH203-[^13^C_6_, ^15^N]-L-leucine ratio (inhibitor-substrate ratio) that ensures comparison close to maximal inhibition.

Nevertheless, the contribution of LAT-1_HD_ on L-leucine uptake also depends on the used L-leucine concentration and the levels of other amino acid transporters present. At low substrate concentrations, only high-affinity transporters have considerable transport velocity. With the elevation of substrate concentration, the high-affinity transporter uptake may reach maximal velocity (v_max_), while the transport velocity of low-affinity transporters increases. Therefore, the contribution on substrate uptake is variable with substrate concentration and further depends on the different levels of transporters, their v_max_, and respective K_m_ (or affinity). Because LAT-1_HD_ is a high-affinity transporter for L-leucine, its contribution to the total L-leucine uptake is maximal at low substrate concentrations. Hence, low L-leucine concentrations are highly beneficial when screening for LAT-1_HD_ inhibitors. However, when using very low substrate concentrations, in addition to high-affinity inhibitors, low-affinity inhibitors will also achieve considerable inhibition and will consequently be identified. 

The reason for the discussed relationships is the dependency of the inhibition extent on the ratio of transporter occupation (binding event) for competitive inhibition. Under the assumption that the transport process kinetic constant is negligible (vastly slower transport than binding kinetics), the extent of inhibition (inhibition/maximal inhibition: I/I_max_) can be approximated by the ratio of inhibitor to substrate concentration, corrected by the inverse ratio of their affinities (dissociation constants of the binding event):
$$\frac{I}{I_{max}}=\frac{1}{1+\frac{K_{D_{I}}}{\left[I\right]}*\left(1+\frac{\left[S\right]}{K_{D_{S}}}\right)}≅\frac{1}{1+\frac{K_{D_{I}}*\left[S\right]}{K_{D_{S}}*\left[I\right]}}\;\mathrm{for}\;\left[I\right]\gg K_{D_{I}}$$
(Derived by inserting the Cheng-Prusoff into the Hill-equation), [S] = substrate concentration; [I] = inhibitor concentration; K_DI_ = inhibitor affinity; K_DS_ = substrate affinity.

Hence, with increasing substrate concentration, inhibition requires increasing values of [I]/K_DI_. This means that at higher substrate concentrations, when using fixed screening concentrations for the potential inhibitors, only higher-affinity inhibitors will achieve considerable inhibition and, hence, will be identified. This characteristic can be used to tailor the outcome of the screening process. For this purpose, setting a threshold value for the extent of inhibition that exceeds the fraction of baseline uptake on total [^13^C_6_, ^15^N]-L-leucine uptake may be reasonable, which ensures that identified inhibitors do actually inhibit LAT-1_HD_.

The identification of a LAT-1_HD_ inhibitor does not allow us to conclude on its specificity, even if the LAT-1_HD_-dependent fraction of L-leucine uptake is the major contribution. A first test for LAT-1_HD_ selectivity can be performed using the [^13^C_6_, ^15^N]-L-leucine assay and methodology. For this purpose, the baseline uptake under maximal inhibition of the identified inhibitor (plateau of the inhibition curve at constant [^13^C_6_, ^15^N]-L-leucine concentration with increasing inhibitor concentrations) can be compared to that of a LAT-1 specific inhibitor (JPH203). If the baseline uptake matches that of the LAT-1_HD_ specific inhibitor, specificity of the identified inhibitor can be assumed. If the identified inhibitor is non-specific, its baseline uptake should undercut that of the LAT-1_HD_ specific inhibitor. This may be performed on hCMEC/D3 cells and additionally on NKIM-6 cells, which show a higher dependency on other amino acid transporters than LAT-1_HD_ for L-leucine uptake. These tests are ideally performed at a very high substrate concentration, which may further increase the contribution of other amino acid transporters on the substrate uptake and can therefore result in a more accurate specificity evaluation.

Further advanced investigations can complement this selectivity evaluation. These would be performed for only a limited set of pre-selected candidates during lead development due to the required high experimental effort. Feasible approaches include a comprehensive set of silencing or overexpression of other amino acid transporters to compare the L-leucine uptake inhibition under these conditions.

The conclusions drawn in this work rely on the fact that JPH203 is really a specific inhibitor for LAT-1 and its mode of action is competitive inhibition. However, the discussed conclusions are generally applicable to any specific LAT-1_HD_ inhibitor that is competitive.

In conclusion, due to the primary dependence of L-leucine uptake on LAT-1_HD_ and the high levels of LAT-1_HD_ expressed, hCMEC/D3 cells are suited for investigations of LAT-1_HD_-mediated transport over the BBB and LAT-1_HD_ inhibitor screening experiments.

## 4. Materials and Methods

### 4.1. Drugs, Chemicals, Solvents, and Materials

The hCMEC/D3 cell-line was purchased from Sigma Aldrich (Taufkirchen, Germany), and NKIM-6 was in our in-house repository [46]. Endothelial growth medium 2 SingleQuots™ and ProSieve^®^ QuadColorTM Protein Markers were purchased from Lonza (Basel, Switzerland). Endothelial Cell Media MV2 was purchased from PromoCell (Heidelberg, Germany). Fetal bovine serum (FBS), penicillin–streptomycin, Dulbecco’s phosphate-buffered saline (PBS), Hanks Balanced Salt Solution (HBSS), trypsin–EDTA solution, 1 M Hepes buffer, GeneElute Mammalian Total RNA Miniprep Kit, [^13^C_6_, ^15^N]-L-leucine, and [^2^H_3_]-L-leucine were obtained from Sigma Aldrich (Taufkirchen, Germany). Heparin was purchased from Biochrom (Berlin, Germany) and geneticin (G418) from PAA (Cölbe, Germany). RPMI 1640 was purchased from PAN Biotech (Aidenbach, Germany). Cell+ culture flasks with vented caps were obtained from Sarstedt (Nürnberg, Germany). Laminin and the following primary antibodies were purchased from Santa Cruz Biotechnology (Dallas, TX, USA): mouse anti-human LAT-1 (D-10) and mouse anti-human β-actin. Phycoerythrin (PE) mouse anti-human CD98 was obtained from BD Biosciences (Heidelberg, Germany). Sheep anti-mouse antibodies linked to horseradish peroxidase and Amersham^TM^ Hybond^TM^ PVDF blotting membranes were purchased from GE Healthcare (Freiburg, Germany). Ammonia solution (28%) was obtained from Merck (Darmstadt, Germany). Acetonitrile (ACN), and formic acid (HCOOH) in the highest analytic purity were purchased from Biosolve (Valkenswaard, The Netherlands). Ultra-purified water was produced with an arium^®^ mini (Sartorius, Göttingen, Germany) ultrapure water system. LAT-1_HD_ inhibitor JPH203 was obtained from Abmole BioScience (Eching, Germany) and BCH from Tocris (Abingdon, UK). RevertAid™ H Minus First Strand cDNA Synthesis Kit, Pierce bicinchonininc acid (BCA)^®^ protein assay, and Pierce ECL Western Blotting substrate were from Thermo Fisher Scientific (Waltham, MA, USA) and the Absolute QPCR SYBR Green Mix from Abgene (Hamburg, Germany). For quantification of *SLC7A5* mRNA, a Quantitect Kit from Qiagen was used (Hilden, Germany). All other primers were obtained from Eurofins MWG Operon (Ebersberg, Germany). Dithiothreitol (DTT), TRIS (2-amino-2-(hydroxymethyl)-propan-1,3-diol), sodium dodecyl sulfate (SDS), Rotiphorese^®^ gel 30, Tween^®^20, TEMED, ammonium peroxodisulfate (APS), and bovine albumin fraction V (BSA) were purchased from AppliChem (Darmstadt, Germany). Protease inhibitors leupeptin and pepstatin were from Biomol (Hamburg, Germany) and pefabloc from Serva (Heidelberg, Germany). Aqueous 1 M NaOH solution was obtained from Roth (Karlsruhe, Germany). Laemmli sample buffer (4×) was purchased from BioRad (Feldkirchen, Germany). 

### 4.2. Preparation of Standard and Quality Control Samples

For [^13^C_6_, ^15^N]-L-leucine, two independent weighings were performed and precisely dissolved in 5 mL H_2_O/ACN (95/5, *v*/*v*) + 0.1% HCOOH in 5 mL volumetric flasks to prepare calibration and QC stock solutions. After 10-fold dilution with H_2_O/ACN (95/5, *v*/*v*) + 0.1% HCOOH, calibration standard spike solutions were prepared at concentrations of 0.5, 1.5, 5, 15, 50, 150, 500, 1500, and 5000 ng/mL corresponding to final concentrations in the sample of 0.1, 0.3, 1, 3, 10, 30, 100, 300, and 1000 ng/mL (cell lysate concentrations of 0.724, 2.17, 7.24, 21.7, 72.4, 217, 724, 2172, and 7240 nM). The second stock solution was used to obtain the QC spike solutions at concentrations of 0.5, 1.5, 1875, and 3750 ng/mL after 10-fold dilution with H_2_O/ACN (95/5, *v*/*v*) + 0.1% HCOOH, which correspond to 0.1 (lower limit of quantification, LLOQ), 0.3 (low QC), 375 (mid QC), and 750 ng/mL (high QC) in the sample (cell lysate concentrations of 0.724, 2.17, 2715, and 5430 nM). [^2^H_3_]-L-leucine stock solution was prepared accordingly and diluted 100-fold in H_2_O/ACN (95/5, *v*/*v*) + 0.1% HCOOH. This sub-stock was further diluted in ACN/H_2_O (95/5, *v*/*v*) + 0.1% HCOOH to obtain the IS spike solution at a final sample concentration of 40 ng/mL. A reference solution prepared at 6 ng/mL [^13^C_6_, ^15^N]-L-leucine and 8 ng/mL [^2^H_3_]-L-leucine in ACN + 0.1% HCOOH was used for system suitability testing.

### 4.3. Cell Culture and Uptake Experiments

NKIM-6 cells were cultivated in Endothelial Cell Media MV2 supplemented with 250 µg/mL G418 and 40 µg/mL heparin. hCMEC/D3 cells were grown in Endothelial Growth Medium 2 containing all supplements provided by the manufacturer (human epidermal growth factors, hydrocortisone, GA-1000, FBS, vascular endothelial growth factor, human fibroblast growth factor B, insulin growth factor-1, ascorbic acid, and heparin). Both endothelial cell lines were grown on Cell+ culture flasks and incubated at 37 °C and 5% CO_2_. Upon reaching 80% confluency, they were passaged 1:5–1:10. Adherent cells were detached by incubation with trypsin–EDTA solution at 37 °C for 2 min. Trypsination was quenched with stop-medium (RPMI supplemented with 10% FBS, penicillin (100 U/mL), and streptomycin (100 µg/mL)). For validation and uptake experiments, 7.5 × 10^4^ cells/cm^2^ hCMEC/D3 and NKIM-6 cells were seeded in 96-well plates coated with 2 µg/cm^2^ laminin and grown until confluent. For validation purposes, cells were directly lysed with aqueous NH_4_OH (10%, 25 µL) in the 96-well plate. For uptake experiments, cells were washed once with warm hepes in HBSS (HHBSS) (100µL/well) before treatment. A 5 mM stock solution of JPH203 was prepared in DMSO. For BCH, a 30 mM stock solution with ultra-pure water was prepared. [^13^C_6_, ^15^N]-L-leucine was diluted in HHBSS for a stock solution concentration of 2 mM. All stock solutions were stored at −20 °C in suitable aliquots to avoid more than one thawing cycle. Dilutions for uptake and inhibition experiments were freshly prepared in 10 mM HHBSS. Either 5 µM JPH203, 500 µM BCH, or the respective dilution of DMSO (for controls) were added to each well. Subsequently, cells were incubated with 0.1–100 µM [^13^C_6_, ^15^N]-L-leucine. For inhibitor experiments, either 0.001–100 µM JPH203 or 0.1–10,000 µM BCH were added, followed by 10 µM [^13^C_6_, ^15^N]-L-leucine. Cells were incubated for 10 min at 37 °C under gentle agitation. Immediately afterward, the [^13^C_6_, ^15^N]-L-leucine solution was removed, the plate was kept on ice, and cells were washed thrice with ice-cold HHBSS. Cells were then lysed with aqueous NH_4_OH (10%, 25 µL) in the 96-well plate.

### 4.4. Sample Preparation for UPLC-MS/MS

After cell lysis with aqueous NH_4_OH (10%, 25 µL), homogenates were spiked with 25 µL IS spike solution directly in the 96-well cell culture plate. To validate samples, 5 µL of the respective calibration or QC solution were added. Subsequently, for protein precipitation, 180 µL ACN + 0.1% HCOOH was added and plates were centrifuged for 10 min at 800× *g*. The 96-well plates were then transferred into the Sample Manager for injection onto the UPLC-MS/MS system.

### 4.5. Instrumental Analysis Parameters

For [^13^C_6_, ^15^N]-L-leucine (138.12 g/mol, ^13^C_6_H_13_^15^NO_2_) and IS [^2^H_3_]-L-leucine (134.19 g/mol) UPLC-MS/MS quantification, a triple-stage quadrupole mass spectrometer (Waters Xevo TQ-XS with Z-spray electrospray ionization (ESI) source) with an Acquity Classic UPLC^®^ (Waters, Milford, MA, USA) was used. Chromatography was performed with a Waters UPLC^®^ BEH Amide column (130 Å, 1.7 µm, 2.1 × 50 mm) at 40 °C. Injection volume was set to 20 µL in partial loop mode, and the flow rate was 0.5 mL/min. Aqueous eluent A consisted of H_2_O/ACN (91/9, *v/v* + 0.1% HCOOH + 0.03% NH_4_OH); the mobile phase B was ACN/H_2_O (95/5, *v*/*v*) + 0.1% HCOOH + 0.03% NH_4_OH. Initially, the eluent composition was 0% A/100% B. After 2.40 min, the percentage of A was linearly increased to 8% A (92% B). A sharp gradient was applied in the next 0.1 min to reach 80% A/20% B at 2.5 min, which was maintained until 2.7 min to flush the column. Then, the initial composition was restored until 2.8 min and kept until the end of the elution at 3.00 min. For mass spectrometry, ESI in positive ion detection mode was used. For quantification, SRM based on CID with argon was used. The parameters were optimized using the auto-optimization feature IntelliStart of the MassLynx V4.2 system software (Waters, Milford, MA, USA) and are shown in Table 3.

### 4.6. Method Validation

UPLC-MS/MS quantification of [^13^C_6_, ^15^N]-L-leucine was validated based on FDA and EMA guidelines for bioanalytical methods [38,39]. In three independent validation runs, the linearity between concentration and measurement, accuracy (measured concentration/nominal concentration), and its precision (SD/mean: % CV) were determined. Selectivity of the measurements was demonstrated by the evaluation of blank cell lysates. All validation batches contained eight calibration samples in duplicates and four QC samples with six replicates each, which were prepared by spiking cell lysate at the respective concentration. The stability of the concentration in the cell homogenates was verified by quantification of QC samples after 24 h storage at 10 °C with freshly prepared calibration samples. In addition, the IS-normalized matrix effect was calculated by dividing the response (peak area ratio of analyte and IS) of QC samples in cell homogenates of blank cell samples spiked after extraction with the corresponding samples in cell-free solvent [52]. The recovery was determined as the quotient of the peak areas of QC samples, and the blank cell samples spiked after extraction with low to high QC concentrations. All samples for matrix effect and recovery were performed in triplicates. Before and after each run, system suitability was established.

### 4.7. Quantification of SLC7A5/LAT-1 mRNA Expression by Real-Time RT-PCR

RNA from cell culture samples of the used cell passages were isolated with the GeneElute Mammalian Total RNA Miniprep Kit following the manufacturer’s instructions. In brief, cells were lysed and thoroughly homogenized with lysis buffer. Homogenates were filtered and RNA subsequently bound to the RNA-binding column. After washing, RNA was eluted with 30 µL elution solution. RNA concentrations were measured with the SpectraMax iD3 (Molecular Devices, Munich, Germany). Subsequently, cDNA was synthesized with the RevertAid™ H Minus First Strand cDNA Synthesis Kit. Expressions of *SLC7A5* mRNA were performed and quantified by real-time RT-PCR with the LightCycler^®^ 480 (Roche Applied Science, Mannheim, Germany), as described previously [53]. Relative expression was determined by normalization to the reference genes *glucose-6-phosphate-dehydrogenase, β-glucuronidase*, and *β2-microglobulin.* Obtained data were evaluated via calibrator-normalized relative quantification with efficiency correction using LightCycler^®^ 480 software version 1.5.1.62 (Roche Applied Science, Mannheim, Germany). Results were expressed as the target/reference ratio divided by the target/reference ratio of the calibrator, corrected for inter-sample variations and technical variance. All samples were performed in technical duplicates, and at least five independent biological samples were measured for each cell line. 

### 4.8. Quantification of Protein Concentration

To normalize the measured [^13^C_6_, ^15^N]-L-leucine concentration in the uptake experiments, the protein content of four wells per plate was determined. For this purpose, wells were lysed with 100 µL Pierce^®^ RIPA buffer containing protease inhibitors (leupeptin 5 μg/mL, pefabloc 1 mg/mL, pepstatin, and aprotinin 1 μg/mL) for 30 min on ice and subsequently centrifugated (4 °C, 15 min 14,000× *g*). The protein concentration of the supernatant was determined by Pierce BCA^®^ protein assay following the manufacturer’s instructions. In brief, BCA reagent was added to the lysates, resulting in a reduction of Cu^2+^ and subsequent reaction with BCA. The amount of emerging purple complex was measured with SpectraMax iD3 after 30 min at 562 nm and is linear to the protein concentration. Absolute values were determined based on a standard curve. To demonstrate the consistency of protein content across the plates, varying positions of wells were analyzed per plate. 

### 4.9. Quantification of SLC7A5/LAT-1 Protein Expression by Western Blot

Cell lysis and determination of protein content were performed as described above (Section 4.8. Quantification of protein concentration). To evaluate the expressional differences between the used cell lines, SLC7A5/LAT-1 was quantified on the protein level by sodium dodecyl sulfate polyacrylamide gel electrophoresis (SDS-PAGE) and subsequent Western blotting. In brief, 15 µg protein mixed with 4× Laemmli buffer containing 1:5 1 M DTT were loaded on 10% polyacrylamide gels. After electrophoresis in tris-glycine running buffer containing 0.1% SDS, the proteins were transferred onto PVDF membranes (70 min, 350 mA) in tris-glycine-transfer buffer containing 20% methanol. Membranes were blocked with 5% BSA in tris-buffered saline with 0.3% Tween 20 for 60 min. Mouse anti-human LAT-1 antibody was diluted 1:200 in blocking buffer (5% BSA) and incubated at 4 °C overnight. The membrane was then washed with Tris-buffered saline containing 0.1% Tween^®^ 20, and anti-mouse secondary antibody conjugated to horseradish peroxidase (diluted 1:2000) was added for 50 min. After washing, immunolabeled proteins were visualized with Pierce ECL Western Blotting Substrates based on chemiluminescence. The substrate was incubated for 1 min, and membranes were subsequently imaged with the Azure 600 imager (Azure Biosystems, Dublin, CA USA) for 10–120 s. For the immunoblotting of β-actin on the same membranes, they were washed with purified water and then stripped by incubation with 1 M aqueous NaOH for 1 min to remove bound antibodies. Subsequently, blots were incubated at 4 °C overnight with mouse anti-human β-actin antibody (diluted 1:2000). After visualization was performed accordingly, LAT-1 band intensities were normalized to β-actin protein levels. Data were generated in three independent experiments. 

### 4.10. Determination of Cell Surface Expression of SLC2A3/CD98 by Flow Cytometry

Expression of SLC2A3/CD98 on HPMEC/D3 and NKIM6 was determined using cell surface antigen staining by flow cytometry. For this purpose, 1 × 10^6^ cells suspended in PBS were incubated for 30 min at 4 °C with a mouse anti-human CD98 antibody conjugated to PE. After washing with PBS containing 2% FBS, samples were measured with a MACSQuant 10 Analyzer flow cytometer from Miltenyi Biotec (Bergisch Gladbach, Germany). Viable cells were first gated in the forward versus side scatter based on size and granularity. A second gate that detected PE-positive cells using a blue laser with an excitation wavelength of 488 nm and a 585/40 nm emission filter was used to determine CD98-positive cells. In each sample, 10,000 cells were measured. Data were generated in three independent experiments using cells from three different passages.

### 4.11. Calculations and Statistical Methods

Waters TargetLynx V4.2 software (Waters, Milford, MA, USA) was used to fit calibration curves based on 1/x^2^-weighted linear regressions of the peak area ratios of [^13^C_6_, ^15^N]-L-leucine and the IS. GraphPad Prism 9 software (version 9.3.1) was used for data visualization, statistical analysis (unpaired *t*-test), and to determine the IC_50_ values of JPH203 and BCH (nonlinear regression, log(inhibitor) vs. response with variable slope (four parameters)). The obtained IC_50_ values were converted to the corresponding inhibition constant K_i_ with the IC-50-to-K_i_ web-based tool [49]. Therefore, the mean of reported K_m_ values for L-leucine uptake by LAT-1_HD_ (32 µM) was used [1,2,50], and classical competitive inhibition was assumed. Microsoft Excel 2010 (Mountain View, CA, USA) was used for general calculations and for the normalization of the [^13^C_6_, ^15^N]-L-leucine cell homogenate amount of substance to protein content [pmol/mg]. Western blot band intensities were quantified with Fiji (Fiji is just ImageJ 2.0.0) [54].

## Figures and Tables

**Figure 1 ijms-23-03637-f001:**
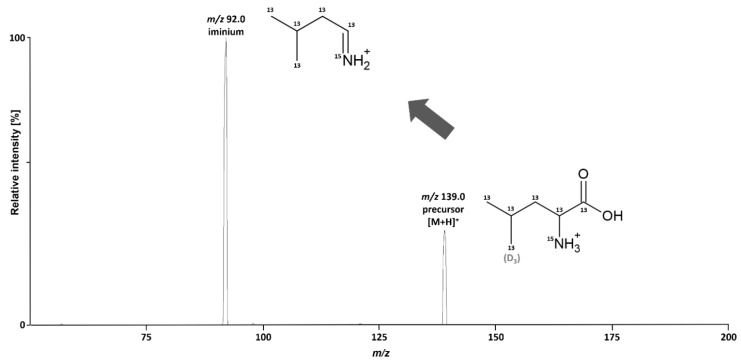
Product ion spectrum of [^13^C_6_, ^15^N]-L-leucine at a collision energy of 9 V.

**Figure 2 ijms-23-03637-f002:**
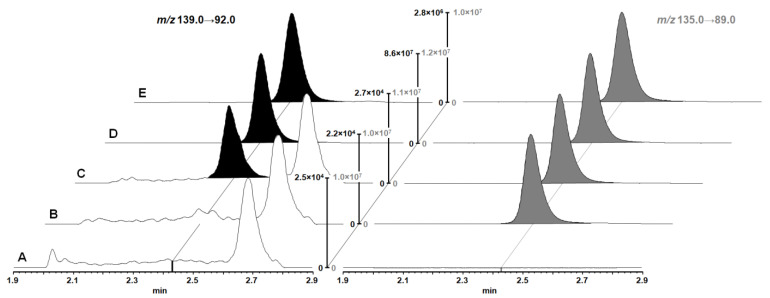
Chromatograms of [^13^C_6_, ^15^N]-L-leucine and IS [^2^H_3_]-L-leucine in cell lysates. The monitored [^13^C_6_, ^15^N]-L-leucine transition is shown on the left in black, and the internal standard (IS) transition is shown on the right in grey. (**A**) Blank sample, (**B**) blank lysate with added IS, (**C**) sample at LLOQ concentration, (**D**) sample at mid QC concentration, (**E**) intracellular [^13^C_6_, ^15^N]-L-leucine concentration after incubation with 10 μM [^13^C_6_, ^15^N]-L-leucine and 100 µM JPH203 for 10 min (quantified [^13^C_6_, ^15^N]-L-leucine concentration: 15.0 ng/mL). Intensities were normalized to the highest signal in the respective run, with the exception of the IS transition of the blank sample, which was normalized to the intensity of the blank sample with added IS.

**Figure 3 ijms-23-03637-f003:**
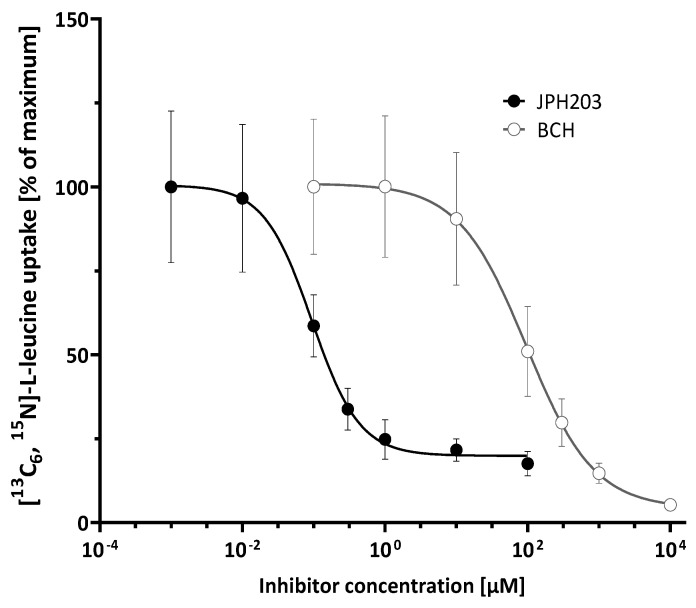
Sigmoidal concentration–response curve of uptake inhibition of 10 µM [^13^C_6_, ^15^N]-L-leucine by JPH203 and BCH in hCMEC/D3 cells. Data are depicted as mean ± SD from three independent experiments with at least three replicates.

**Figure 4 ijms-23-03637-f004:**
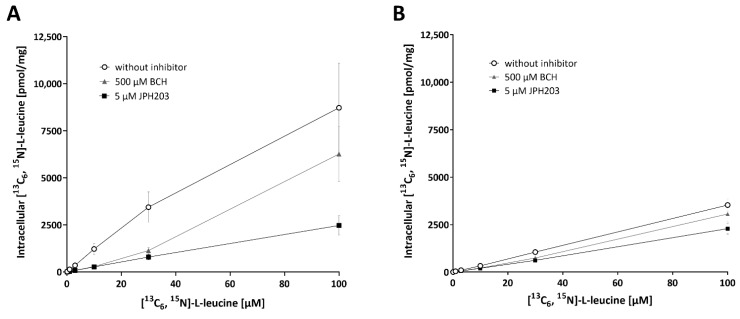
Concentration-dependent uptake of [^13^C_6_, ^15^N]-L-leucine without inhibitor, with 5 µM JPH203 and 500 µM BCH in (**A**) hCMEC/D3 and (**B**) NKIM-6. Concentration of intracellular [^13^C_6_, ^15^N]-L-leucine was normalized to protein content. Data are depicted as mean ± SD from three independent experiments with at least three replicates.

**Figure 5 ijms-23-03637-f005:**
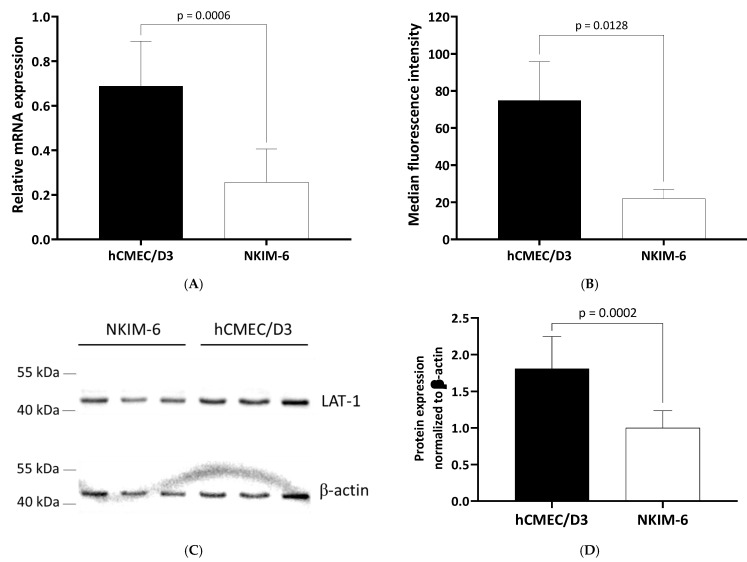
Expressional differences of *SLC7A5* and SLC2A3/CD98 between hCMEC/D3 and NKIM-6 cell lines. (**A**) Normalized levels of *SLC7A5* mRNA. Data are the mean ± SD of at least five biological replicates per cell line with two technical replicates each. (**B**) Median fluorescence intensity measured by flow cytometry of cells stained with PE-conjugated mouse anti-human CD98 antibody. Data are the mean ± SD from the median of three independent experiments. Statistical analyses were performed with unpaired *t*-tests. (**C**) Representative immunoblot of LAT-1 in NKIM-6 and hCMEC/D3. Bands of LAT-1 and β-actin are seen at the expected molecular weight of 40 and 42 kDa, respectively. For each cell line, three cell passages are shown: for NKIM-6 passages 50, 52, 53, for hCMEC/D3 32, 28, 33 (from left to right). (**D**) Quantitative analysis of three Western blots. Arbitrary units of LAT-1 bands were normalized to corresponding β-actin bands. Data are shown as mean ± SD of fold changes to the mean of the NKIM-6 values.

**Table 1 ijms-23-03637-t001:** Matrix effect and recovery.

	Low QC	Mid QC	High QC	IS
	0.300 ng/mL	375 ng/mL	750 ng/mL	40 ng/mL
Matrix effect IS normalized [%]	90.5	106.6	105.1	-
Recovery [%]	94.6	95.8	97.9	101.1

IS: internal standard, QC: quality control. N = 3 replicates at each QC concentration.

**Table 2 ijms-23-03637-t002:** Quality control results for functional LAT-1_HD_ assay.

		LLOQ	Low QC	Mid QC	High QC
		0.100 ng/mL	0.300 ng/mL	375 ng/mL	750 ng/mL
Intraday					
1	Mean [ng/mL]	0.110	0.323	406	794
	Accuracy [%]	110.0	107.8	108.3	105.9
	Precision [% CV]	6.43	5.26	5.43	3.73
2	Mean [ng/mL]	0.103	0.307	399.0	796
	Accuracy [%]	102.5	102.2	106.4	106.1
	Precision [% CV]	4.88	4.86	1.42	5.06
3	Mean [ng/mL]	0.100	0.307	384	743
	Accuracy [%]	100.0	102.2	102.5	99.1
	Precision [% CV]	11.0	4.07	2.38	0.79
Interday					
	Mean [ng/mL]	0.104	0.312	397	778
	Accuracy [%]	104.2	104.1	105.7	103.7
	Precision [% CV]	4.08	2.52	2.27	3.16

CV: coefficient of variation; LLOQ: lower limit of quantification; QC: quality control. N = 6 replicates at LLOQ and each QC concentration.

**Table 3 ijms-23-03637-t003:** Optimized parameters for the MS/MS detection of [^13^C_6_, ^15^N]-L-leucine and [^2^H_3_]-L-leucine in positive heated ESI and SRM.

Parameter	[^13^C_6_, ^15^N]-L-Leucine ([^2^H_3_]-L-Leucine)
Capillary voltage	0.5 kV
Cone voltage	26 V
Cone gas flow	150 L/h
Source temperature	150 °C
Desolvation gas flow (N_2_)	1000 L/h
Desolvation temperature	600 °C
SRM transition [*m*/*z*]	139.02 → 92.0 (135.0 → 89.0)
Dwell time	80 ms
Collision energy	9 V
Collision gas flow (Ar)	0.15 mL/min

ESI: electrospray ionization, SRM: selected reaction monitoring.

## Data Availability

The data are available from the corresponding author upon reasonable request.

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
