# Peer review of "Functional Characterization of the Solute Carrier LAT-1 (SLC7A5/SLC3A2) in Human Brain Capillary Endothelial Cells with Rapid UPLC-MS/MS Quantification of Intracellular Isotopically Labelled L-Leucine"

_ijms, 2022, doi:10.3390/ijms23073637_

Round 1

Reviewer 1 Report

The manuscript describes the functional characterization of LAT-1 in brain capillary endothelial cells with LC-MS/MS quantification of intracellular isotopically labelled L-leucine. According to the authors, the manuscript describes the partial validation of the novel LC-MS method to support CNS drug-delivery studies based on LAT-1 targeting. Authors also investigate the IC50 of JPH203 and BHC on hCMEC/D3 cells. In addition, isotopically labelled L-leucine uptake was determined on two different cell lines. These studies further validate the new LC-MS/MS method to be fit for the purpose.

Although I found many items in the manuscript sufficiently well designed and executed, I still ,unfortunately, believe that the manuscript not fully fits in to the scope of the International Journal of Molecular Sciences.

A few specific comments include:

  • Abstract: FDA and EMA guidelines standardize validation related parameters and are therefore appropriate references. The manuscript would be better not to mention those in the text. Could the authors refer to them by reference numbers only? Instead, it would be beneficial to describe which validation parameters are studied during the validation step. In addition, I consider term “validation” quite useless in scientific text. Methods should always be validated (and be described in adequate level to show validity/fit for purpose of the method for the purpose).
  • Abstract: BCH is not LAT-1 specific substrate, like the authors point out in Introduction and Chapter 3.4 (page 8, line 278). Please rephrase the sentence in abstract (lines 23 and 24).
  • Introduction: Please, could the authors identify other transporters of L-leucine. Are there any other transporters involved to intake of L-leucine and how specific is LAT-1 transportation?
  • Chapter 2.1 (page 3, line 98): Please, could you use HCOOH as an abbreviation for formic acid.
  • Chapter 2.2 (page 3, line 117): mid QC level at 60 ng/ml, but in Table 3 Mid QC is 375 ng/ml. Please, could the authors correct these values.
  • Chapter 2.2 (page 3, line 115): The concentration gap between two highest calibration levels seems to be quite big. Have the authors considered to add one calibration level between two highest concentrations? What is the dynamic range of the calibration?
  • Chapter 2.5 (page 4, line 151): The injection volume is very high for the column used (2.1 * 50 mm). Please justify this in the chapter 3.1 (Chromatography and mass spectrometry).
  • Chapter 2.5 (page 4, line 154): The gradient of the method seems very fast. I wonder, if there are enough time for compounds with high retention time to eluate from the column. Could authors speculate this issue in chapter 3.1 (Chromatography and mass spectrometry). Combination of large injection volume to inadequate washing step of mobile phase gradient could lead to the systematic errors and issues with selectivity of the results.
  • Chapter 3.1 (page 6): I found the chapter 3.1 a bit short, and it could describe more broadly the observations made during the method development. Could the authors describe for example columns and mobile phases tested. In addition, all findings related to sampling and sample preparation is vital for the method. Most of the result error is caused by steps before to instrument.
  • Table 1: L/Hr or L/h, please harmonize.
  • Chapter 3.1 (page 6, line 239): please, write in italics m/z (and same in centrifuge related g).
  • Table 3 and Figure 4: Consider using same concentration units throughout the whole manuscript, either ng/ml or µM.

Reviewer 2 Report

General Comments

Quite a nice, robust method-driven paper. The method itself is well described and validated in terms of the UPLC-LC/LC quantification etc. However, three main problems which require attention need to be dealt with before the paper is ready for publication. Firstly, the referencing in the introduction needs substantial adjustment (and possibly re-writing to go with it) – see individual comments below. Secondly, more figures are required to accompany the validation and quantification Tables a and 3 presented in the results, along with further explanation in the text describing what was done (refer to the methods if appropriate) and how you interpret the results. Thirdly, The authors need to address (not necessarily with further results but they may do or, alternatively, incorporate answers in the discussion) the fact that BCH in particular is not a specific inhibitor of LAT1.

Introduction and Methods

  1. Lines 44-45: the phrase ‘CD98 contributes to the transport function as a chaperone’ is a contradiction in terms; if CD98 is only a chaperone of LAT1 this molecular function isn’t, by definition, a ‘transport function’. Transport function refers specifically to the enzyme/transporter's activity at the Plasma membrane in translocating substrate from one side of the membrane to the other. Please re-phrase and see next comment.
  2. On the point that LAT functions at the plasma membrane independently of CD98 or not: this point is controversial. The recent structure paper of the LAT1-4F2hc showed biochemical evidence that the heavy chain 4F2hc (CD98) is required for the transport function at the membrane i.e. the opposite of the information provided here. The recent reviews provide an excellent overview of the evidence for and against the essentiality of 4F2hc for the transport function. Please re-word the text and cite additional relevant primary papers to reflect that this is still an unclear finding.
  3. On reference 4, where it is used in line 46: the paper cited is not the most appropriate, it is not the only paper which contributed to the elucidation of residues involved in binding. It was proceeded by many and rather confirmed much that was already known. As with the previous points on appropriate, accurate and comprehesive referencing this review10.1007/5584_2020_584 should be consulted and itself cited to alert the reader to where a comprehesive overview of substrate binding can be read and to cite specific relevant papers which were key to understand substrate binding in LAT1. Also, a minor note, of you consult the review and specific research papers you will see that binding site of LAT1 also involves more residues other than those in TM1 and TM6 as stated in you introduction.
  4. Line 49-50: While it is not incorrect to say that LAT1 supplies leucine and other essential amino acids (AA) to cells, it is not entirely clear from the BBB or in other tissues whether LAT1-4F2hc is responsible for the net accumulation of amino acids and whether is it the major accumulator. See this paper DOI: 10.1038/s41467-021-25563-x PMID: 34489418, for some understanding of the complex role LAT1-4F2hc plays in the net accumulation of essential amino acids. See also, this review PMID: 33606172 DOI: 10.1007/s11064-021-03261-w for a good summary of the evidence for LAT1-4F2hc role in net amino acid accumulation across the BBB. Without proscribing exactly what should be written here, the authors do need revise these passage to illustrate the various ways LAT1-4F2hc has been shown to function in co-ordination with other transporters to achieve net AA accumulation e.g. ASCT2 (slc1a5), SNAT2 (slc38a2). After all, as an isolated transporter, LAT1 is an obligatory exchanger with no secondary energy source and relies solely on the electro-chemcial gradient of its amino acid substrates, so it could do nothing by itself than equilibrate substrate across the membrane. The review and paper should be cited and any other relevant primary research papers outlined in the review that illustrate these points also cited.
  5. Also, on the same point as above; what is the evidence that histidine may be a antiport substrate predominately driving net essential AA accumulation such as Leu? Sure, His is a substrate of LAT1, but so are several other abundant intracellullar amino acids. So, without a citation of evidence in the form of research papers (not just a review) in this sentence, the assertion merely reads as speculation and not established. Please consult the reviews I have provided (one of which you have cited already) to see if specific evidence is provided for you assertion and if not, and as with the previous point, what is known about LAT1’s role in the net accumulation of what amino acids.
  6. Lines 55-57: As with above comments, please ensure that the references cited in relation to LAT1 over-expression in various cancers are comprehensive. LAT1 has been the subject of at least a dozen research papers in gliablastomas alone demonstrating over-expression and treatment, diagnostic and treatment potential. If you are going to cite more than just the first research paper for each over-expression finding you report why not cite them all? Alternatively if you are making specific points, i.e. over-expression in a particular cancer type, why is not simply the first primary research paper to establish this fact enough? I do not suggest that you cite all these research papers on LAT1 and various cancers ad nauseum (nor that you should even cite the general reviews) just that you should tidy the referencing in the section up to be specific for the point being made, or, if you want to be more comprehensive, to be fully comprehensive and cite all relevant papers, not just 2 in each instance.
  7. Lines 64-65. I can update your knowledge. Several recent papers have utilised various tandem chromatography-mass spectrometry and fluorescence imaging based methods to study LAT1-4F2hc function in various expression/cell systems. Please modified this sentence to reflects this and cite papers in this regard.

It may be also worth detailing when citing these papers the specific technique used in each case to study LAT1-4F2hc function.

As I see it, the novelty in your method is that it is the first to utilize UPLC-MS/MS to studying LAT1 function in BBB models, the first to do accurate quantification using different isotopologues, and the upscaling to a 96-well UPLC-MS/MS screening format. More specific that the current claim to novelty but still very much worth highlighting.

8 A nice comprehensive method section.

Results and Discussion

  1. The authors need to present results or discuss in depth the fact that BCH inhibits all so-called ‘systems L’ transporters of amino acids (LAT1, LAT2, LAT3, LAT4) and the other AA transporter B0AT1 (slc6a19) and ATB0,+ (slc6a14)

The authors must address specifically either through addition results or extensive discussion, a) are these other transporter expressed to significant degrees in there cell cultures? b) how the BCH results are likely to be re-interpreted if they are? and c) how they might incorporate specificity test for inhibitors into their method, especially as, according to their own rationale, they are developing a method for future screening of potential inhibitors. The authors mention in the current discussion that BCH is non-selective but do not address these very important concerns. Note, I am not suggesting that the non-selective nature of BCH invalidates any of there results, on the contrary, clearly the major aim was to develop an alternative method to detect transporter inhibition and their results are independent of the transporters inhibited. However, it is important to acknowledge this limitation and discuss in depth how it might be over come to ensure selectivity of potential inhibitors in the context of their assay.

  1. Line 238: Is it possible for the authors to show in a figure more data on the CID optimised CID they mention (not just the Fig. 1 at 9V. Did they scan a range of voltages and did the increase result in a proportional increase in fragmentation and appearance of the iminium ion (or decrease if they lower the voltage)? It would, for sake of comprehensiveness, be nice to see the full set of EICs for various voltages and the fragment ions that appear (ir at least report them).
  2. Lines 244-245: Figure 2 doesn’t really relate to the text you have provided for it. The text suggests a figure demonstrating optimisation of the HILIC column type and mobile phase elution profile yet in the actual figure a series of the TICs for both Isotopologues of L-Leu fro different cell lysates and control samples. Either show a new figure demonstrating the optimisation of the column and elution profile (as is stated) or remove the reference to figure 2. The actual data for figure 2 is mention in the next sentence.
  3. A general point for the results. You need to cut down on, or at least give the full name when first mentioned, of the abundant use of MS and UPLC acronyms in the results. Either introduce them at first usage in the results by there full name (with the acronym in brackets after) or give some of the less commonly used ones as their full name on each usage. If this was a specialist chromatography journal (e.g. analytical chemistry, Journal of Chromatography) I would not ask for this but IJMS is more a general molecular science readership.
  4. The calibration and quantification validation of the L-Leu [ 13C6 , N15]-leucine isotopologue needs to be shown as a figure(s). Please show the calibration curve referred to with the fitting and pearson's R2 and also for the ‘assay’ you mention. It is unclear what assay you are measuring – is this an uptake assay of the isotopologue followed by the UPLC-MS/MS quantification as is described in the method? Please make this section clearer by outlining exactly what assay is being conducted. If this is the uptake assay, where is a figure showing the uptake in the presence of L-Leu isotopologue vs the non-incubated cell lysates? In fact the way you have written section 3.4 is a good example of what is required to evaluate section 3.3. This section needs some work and further figures.
  5. Please explain how the FDA and EMA guidelines are validated in this section with specific reference to the data generated. I know these standards are outlined in the methods but the results section 3.3 is not understandable according to your own validation criteria without specific reference to standards within these guidelines and explicit mention of how you data meets them.
  6. Reference 46 is not appropriate for the information you are providing, it is a review. The correct papers to cite (where IC50 or inhibition data for BCH and JPH203) should be looked up and cited.
  7. Line 286-288: Not exactly true nor accurately explained. The relationship between IC50 and Ki is complicated but the idea that IC50 is not an accurate reflection of the thermodynamic constant Ki is true only under certain circumstances see https://doi.org/10.1021/ed080p214 for example. Please correct to be more specific and give a rationalisation of why you specifically wish to convert your data into Ki and why this was valid for the data obtained.
  8. Just out of curiosity, where you able to detect BCH incubated cell lysates from you experiments? It is to my knowledge, and as you mention, transported by LAT1 and other transporters.
  9. Section 3.5 is mislabelled. It currently reads ‘3.3’ , hence currently labelled section 3.5 should also be section 3.6.
  10. There is a strange steak of chemiluminescence signal across figure 5C the lower panel. What is this?

Reviewer 3 Report

The manuscript established a 96-well format based semi-high throughput UPLC-MS/MS assay for stable isotopically labelled leucine ([13C6, 15N]-L-leucine). And this assay was applied to investigate the IC50 of two LAT-1 inhibitors (JPH203 and BCH) on hCMEC/D3 cells and uptake experiment on two human capillary endothelial cell lines (NKIM-6 and hCMEC/D3). The experiment was solid with sufficient data supported and the result and conclusion are quite straight forward.

  1. As this method considered only one transition in SRM, whether other potential transitions have been considered or not? If so, could the author address this information in the supplementary information for further reader interests.
  2. One simple protein structure of LAT-1HD could be considered in both the introduction section and the result section.
  3. The autosampler temperature information was lost. And if it is 10 ℃, please address this with stability validation.
  4. Figure 2B showed two tiny little peaks at the retention time of [13C6, 15N]-L-leucine, did this tiny peak indicate potential IS compound influence? If so, did the author eliminated as background? And is this tiny peak stable? And has the author considered other transitions to avoid this issue?
  5. The cell passage information could consider added on Figure 5C.

Round 2

Reviewer 1 Report

The manuscript has improved considerably and to my comments had been answered to a sufficient extent.

Author Response

We thank the Reviewer for the kind evaluation.

Reviewer 2 Report

General Comments

The authors have satisfied most of concerns and corrected sufficiently the previous manuscript. There remain 2 original points which have not, however, been addressed entirely satisfactorily which I have detailed below.  These are however, relatively minor points.

Specific Comments

  1. Response to Author (original Introduction comment 7, Lines 111-113 in manuscript v.2): Your response to my comment is simply not true: at least one of the papers I mentioned in my initial comment use 13C (i.e. a non-radioactive isotopologue DOI: 10.1038/s41467-021-25563-x) and several more, if not all of them, are measuring the direct uptake of leucine (or other substrates) via the transport simply by measuring the baseline cytosolic concentration measurement before application of substrate and/or taking account the starting endogenous concentration of leucine as well as other transporters of leucine. Indeed, the example paper I cite does this quantification in quiet a sophisticated and comprehensive manner using a combination of computational metabolism, identification of all transporters for all amino acid substrates present in a cell line, and taking account of metabolism and using 13C-labelled isotopologues of amino acids including leucine. Other papers I referred the authors to, measure 12C leucine (or naturally occurring isotopic abundances) in systems (e.g. oocytes) where the background metabolism and endogenous concentrations have been well account for and subtracted from. Furthermore, such heterologous expression systems as X.laevis oocytes, S.cerevisiae, or insect cells, over-express one specific transport such that the S:N is many fold larger than for cells exhibiting endogenous expression levels. If the authors’ objections to my initial comment is that these papers are not studying LAT1-4F2hc, then this too is incorrect – all of the papers I cited were directly studying LAT1-4F2hc function as part of larger studies. I encourage the authors to have a look at these papers in depth and they will see that my assertion is correct. I also did not mention that the papers I referred the authors to were simply measuring Leu concentration, they are all, contrary to this assertion, aiming to and, in most cases successfully, measuring LAT1-4F2hc activity directly - many of them in competitive, physiologically mimetic environments. I acknowledge, the other changes made reflecting the actual novelty of the paper, I do request that this introduction sentence also be modified to reflect the information I have provided and referred the authors to.
  2. Response to Author (original Results and Discussion comment 1). I think the authors have misunderstood, somewhat, my criticism of the manuscript on this point. I accept the author’s responses that clarify and agree that the results do establish the cell lines as BBB models for studying LAT1 inhibition. I am well aware that JPH203 is a specific inhibitor of LAT1 in contrast to BCH. The specific question I wanted answered and addressed here is how it will be determined that future potential inhibitors are specific for LAT1? In other words, my comments and questions referred to BCH as an example of a potential future identified inhibitor of Leu uptake in these cell models and the interpretation of these results in a cell systems which may contain multiple Leu transport pathways. The essence of the issue concerning the presence of multiple transporters is stated by the authors themselves in response (added to the text, helpfully) that L-Leu accumulation was reduced 77% in hCMEC/D3 cells at 1µM Leu. The problem with this emphasis is it does not represent the physiological (serum) L-Leu concentrations (which are typically 150-200 uM/L see Plasma concentrations and intakes of amino acids in male meat-eaters, fish-eaters, vegetarians and vegans: a cross-sectional analysis in the EPIC-Oxford cohort (nature.com) for one of many examples that could be cited) under which LAT1 would be inhibited by JPH203 in vivo. Nor in your figure 4A is the JPH203-dependent inhibition of LAT1 leu uptake actually 77% at [Leu] greater than this small value - it is far lower and increasingly so as the concentration of Leu increases. The focus on inhibition at 1uM Leu is neither the most relevant Leu concentration nor is it particular visible on the graph. Indeed at 100 uM Leu (starting to approach a physiological relevant serum concentration) the JPH203 specific inhibition is ~30% of the total Leu accumulation. This later figure is the more relevant to emphasize in the manuscript text and, furthermore, illustrates the importance of my initial comment concerning BCH and the presence of multiple Leu uptake pathways. The model, as stated now several times and as you point out in your response to the first draft, is not invalidated as a cell line to study LAT1 inhibition but it does raise questions as to how it will be determined that future potential inhibitors are specific for LAT1? If a new inhibitor, picked up in screening assays, reduced L-Leu in hMEC cells by 80% at 100 uM [Leu] (or conversely by only 10%) how are you going to distinguish, using this assay or others, what component of that Leu uptake inhibition are due to LAT1 and what component die to other Leu transporters? In other words, how do narrow or remove the identification of false positive and negatives based on misappropriated specificity in future inhibitor screens? This question and my concerns would be relevant for all future use of this BBB cell model to identify LAT1 specific inhibitors.

I apologise if the focus of my initial comment was not clear – I am loath in the review process to request specific experiments or ask authors to address specific questions if the results, as they stand, are not actually invalid (which yours are not). This is because it cannot often be difficult to know how capable or pragmatic such requests could be dependent of expertise and equipment availability. I appreciate the authors have already added text which clarifies multiple targets for BCH but they must address the above question and as asked in the comments to the first version: how they might incorporate specificity test for inhibitors into their method, especially as, according to their own rationale, they are developing a method for future screening of potential inhibitors ? According to the concerns I have outline above. You might begin to think about and discuss using, for example, competition assay with different amino acids to distinguish transporters and their contribution to Leu uptake inhibition in future drug discovery, or what complementary methods might be used to overcome the specificity limitation of the current assay. I think I have been quite reasonable in not asking for further experiments that ask the authors to prove the validity of the cell model to test for LAT1 specific inhibitors, which is the stated aim of the paper. I simply ask them to think further about the points I have raised and provided further discussion, explanation and clarification. After all, any further development of these cell lines to test LAT1 inhibitors will continue to come up against the same queries.
